# Biocompatibility and Mechanical Stability of Nanopatterned Titanium Films on Stainless Steel Vascular Stents

**DOI:** 10.3390/ijms23094595

**Published:** 2022-04-21

**Authors:** Cagatay Yelkarasi, Nina Recek, Kursat Kazmanli, Janez Kovač, Miran Mozetič, Mustafa Urgen, Ita Junkar

**Affiliations:** 1Department of Metallurgical and Materials Engineering, Istanbul Technical University, 34469 Istanbul, Turkey; yelkarasi@itu.edu.tr (C.Y.); kursat@itu.edu.tr (K.K.); 2Jozef Stefan Institute, Jamova Cesta 39, Sl-1000 Ljubljana, Slovenia; nina.recek@ijs.si (N.R.); janez.kovac@ijs.si (J.K.); miran.mozetic@ijs.si (M.M.)

**Keywords:** cardiovascular stents, nanopatterning, plasma treatment, endothelium cells, smooth muscle cells, cell viability, hemocompatibility, restenosis

## Abstract

Nanoporous ceramic coatings such as titania are promoted to produce drug-free cardiovascular stents with a low risk of in-stent restenosis (*ISR*) because of their selectivity towards vascular cell proliferation. The brittle coatings applied on stents are prone to cracking because they are subjected to plastic deformation during implantation. This study aims to overcome this problem by using a unique process without refraining from biocompatibility. Accordingly, a titanium film with 1 µm thickness was deposited on 316 LVM stainless-steel sheets using magnetron sputtering. Then, the samples were anodized to produce nanoporous oxide. The nanoporous oxide was removed by ultrasonication, leaving an approximately 500 nm metallic titanium layer with a nanopatterned surface. XPS studies revealed the presence of a 5 nm-thick TiO_2_ surface layer with a trace amount of fluorinated titanium on nanopatterned surfaces. Oxygen plasma treatment of the nanopatterned surface produced an additional 5 nm-thick fluoride-free oxide layer. The samples did not exhibit any cracking or spallation during plastic deformation. Cell viability studies showed that nanopatterned surfaces stimulate endothelial cell proliferation while reducing the proliferation of smooth muscle cells. Plasma treatment further accelerated the proliferation of endothelial cells. Activation of blood platelets did not occur on oxygen plasma-treated, fluoride-free nanopatterned surfaces. The presented surface treatment method can also be applied to other stent materials such as CoCr, nitinol, and orthopedic implants.

## 1. Introduction

According to the World Health Organization [1,2], cardiovascular diseases represent the number one cause of death globally, accounting for almost one-third of all cases. The number of coronary bypass surgeries, a radical way of treating cardiovascular diseases (CVD), decreased dramatically with the invention and application of coronary stents. Today, the most widely used cardiovascular stent materials are 316 LVM stainless-steels, CoCr alloys, and nitinol [3]. In the early decades, bare metallic stents (*BMS*) were successfully used for the treatment of more than 2 million people each year [4], even if they may cause other complications such as re-blockage of the vessels, also known as in-stent restenosis (*ISR*). *ISR* is widely observed in 20–30% of the patients six months after the stent implantation [5]. This post-implantation complication is caused by vascular smooth cell migration, proliferation, and growth of the arterial inner wall, and can be described as “an exaggerated healing process” because of the insufficient biocompatibility of the stent material [6]. In order to decrease the *ISR* rate, drug-eluting stents (*DES*) were developed by coating the bare metal stents with immunosuppressive or cancer treatment agent-containing polymeric materials to inhibit vascular smooth cell growth [7]. However, long-term research showed that the risk of stent-induced thrombosis rate of *DES* is much higher than *BMS* [8,9,10].

Additionally, there was always a risk of peeling off the polymer coating during stent deployment [11]. The new generation of *DES* with a bioabsorbable polymeric coating showed promising results compared to the first-generation *DES* [12,13]. However, it is still quite challenging to significantly reduce the restenosis rate without using drugs such as sirolimus or paclitaxel.

Biocompatibility of the implant surfaces can also be improved with chemically stable inorganic material coatings [8,14]. Metal oxides such as TiO_2_ are extensively used for this purpose [15,16], as they exhibit good biocompatibility and can be produced in various morphologies, including nanostructures. Many studies have already shown that nanostructured surface features significantly influence biological responses and can even be selective for specific cell types [17,18,19,20,21]. Nanostructures can be fabricated by various methods, such as sandblasting [22], hydrothermal processes [23,24,25,26], anodization [27,28], or surface-alloying followed by selective etching [10].

Karpagavalli et al. [29] reported that nano-topography produced by deposition of nanostructured TiO_2_ onto Ti-alloy surfaces enhanced biocompatibility by decreasing the human aortic smooth muscle cells’ integration. Nasakina et al. [30] studied the biocompatibility of nanostructured nitinol (NiTi: 55.91 wt.% Ni–44.03 wt.% Ti) coated with titanium or tantalum layers produced with magnetron sputtering. They have found that coated samples exhibited higher mitotic activity than the NiTi reference sample with the formation of a merged-cell monolayer of the myofibroblasts and the mesenchymal stromal cells. Another study by Chen et al. [31] showed that mesenchymal stem cells (*MSCs*) proliferated better on the nano-hydroxyapatite (*HAp*) coating compared to bare titanium or micron-sized hydroxyapatite-coated surface, indicating that the *HAp/Ti* nanocomposite has good biocompatibility and bioactivity for orthopedic and dental implant applications.

For implants in contact with blood, such as vascular stents, inhibiting or decreasing the potential of thrombosis, associated with high platelet adhesion and its activation on the implant surface, is a primary concern. The adhesion of platelets is an indicator of surface hemocompatibility: the lower the platelet adhesion and their activation on the surface, the higher the material biocompatibility with blood [32]. It has already been shown that appropriately nanostructured surfaces could influence platelet adhesion [33,34]. This improvement was ascribed to altered conformation of adsorbed proteins on the surface, mainly fibrinogen, which was recognized as an essential factor determining platelet adhesion and activation. It was shown in a recent study by Firkowska-Boden et al. that confirmation of fibrinogen molecules changes due to the altered surface nano-topography, which further dictates platelet interactions [33]. However, it should be emphasized that other physicochemical properties of the material’s surface, such as chemical composition and wettability, play an essential role in blood platelet interactions [34,35]. Thus, plasma treatment may also be used for fine-tuning the nanostructured surface properties to improve biocompatibility. This treatment can be performed using various gaseous discharges to make surfaces either hydrophilic or hydrophobic and induce different chemical surface compositions, surface charge, roughness, and crystallinity [36]. These factors have an essential role in immobilizing proteins and cells [37,38,39]. An excellent review of plasma treatment and cell adhesion is provided in a paper by Griesser et al. [40].

In the studies aiming to investigate the contribution of brittle oxide-based nanostructures such as anodic oxides to the biocompatibility of coronary stents, the nature and integrity of the treated surface and its performance during implantation are not afforded sufficient consideration. The stents are plastically deformed during implantation, which increases the brittle top layer’s cracking and delamination risk. Thus, ceramic coatings on stents, such as titanium oxide, have risks because of the high plastic deformation during their expansion. As a result, cracked or delaminated coating layers may detach from the surface under mechanical stresses, including forces accompanied by blood flow or cyclic expansion of the blood veins. This phenomenon not only damages the desired surface chemistry and nano-topography, but may also increase the risk of restenosis and other complications arising from particle debris [41]. In an ideal case, a coating on a *BMS* should be as flexible as the metallic substrate itself, and at the same time as biocompatible as a nanostructured film. Fabricating such coatings or surface modifications that exhibit these properties still presents a scientific and technological challenge.

In this work, a unique surface modification method is reported for producing a highly biocompatible nanopatterned surface that is expected to preserve its integrity during plastic deformation of the stents. The method briefly consists of deposition of titanium on AISI 316 LVM stainless-steel stent materials, followed by anodization and removal of the nanoporous anodic oxide from the substrates by ultrasonic cavitation, leaving behind a nanopatterned metallic titanium layer on the SS. The samples are also subjected to an additional treatment with non-equilibrium gaseous oxygen plasma for improving biocompatibility. The mechanical integrity of nanoporous anodic oxide and nanopatterned metallic titanium-covered surface layers during plastic deformation was also investigated. Furthermore, in vitro studies with human coronary artery endothelial cells (*HCAEC*) and human coronary artery smooth muscle cells (*HCASMC*) and whole blood were performed to evaluate the influence of surface modification on the biological response. The innovative, nanopatterned, metallic titanium-coated stainless-steel surfaces have shown superior biocompatible properties and, at the same time, good coating stability, which is of primary importance for its application for vascular stents.

## 2. Results

A schematic illustration of procedures used to prepare the nanopatterned surfaces is presented in Figure 1. Accordingly, after surface polishing of stainless-steel (SS), a four-step procedure was applied:Step #1: Titanium coating on SS (*pTi*).Step #2: Partial anodization of titanium-coated SS (*aTiO_2_*).Step #3: Removal of anodic porous anodic oxide with ultrasonication, leaving behind a thin nanopatterned Ti layer on SS (*cTi*).Step #4: Brief oxygen plasma treatment (*cTi+P*).

In addition to the steps above, *pTi* and its plasma-treated version (*pTi+P*) were used as a reference in the biological tests.

### 2.1. Optimization of Anodization Duration

During the anodization of *pTi*, the thickness of the metallic titanium that will remain on the SS after ultrasonication is directly related to the anodization time. Therefore, we first anodized the entire titanium film on SS and recorded the anodization current versus time. With the initiation of anodization, the current sharply decreased, indicating the formation of a barrier titanium oxide layer, followed by a slight increase in current with the formation of pores. After the pore formation, the current decreased monotonously with the increasing thickness of the oxide film. With a further increase of anodization time, the current started to increase, with the beginning of the anodic dissolution of the SS surface (Figure 2). Accordingly, the time needed for anodization of the entire titanium film was determined as 500 s. Thus, for the anodization of approximately 500 nm of the metallic titanium on the SS substrate, we anodized the sample for 250 s. The validity of the method was verified by measuring the remained titanium layer thickness by XRF after removal of the anodic oxide with ultrasonication. These measurements showed that the thickness of the remnant metallic titanium film was in the range of 500 ± 40 nm. We used this validated methodology to prepare the samples for further testing.

### 2.2. Morphological Characterizations

Morphological changes after accomplishing each step were examined by scanning electron microscope (SEM) (Figure 3). The SEM image of the 316 LVM stainless-steel sample after titanium film deposition (Step #1) is presented in Figure 3a. A relatively smooth surface was observed (*pTi*). Such a surface is typical for deposition processes using magnetron sputtering with moderate biasing of the substrate.

Figure 3b shows the SEM image after the anodization process (Step #2). The titanium layer was transformed into a nanoporous oxide structure (*aTiO_2_*). The lateral dimensions of the pores were about 100 nm.

The ultrasonication (Step #3) enabled the removal of the nanoporous oxide film and revealed the nanopatterned titanium layer with a hemispherical nanostructure (*cTi*) (Figure 3c).

The SEM image of the boundary line of the ultrasonicated area clearly shows that the process successfully removed the brittle, weakly adhered nanoporous TiO_2_ without causing a significant modification of the remnant nanopatterned metallic titanium layer (Figure 4).

The depth and diameter of the nanopatterns were measured with AFM. Figure 5a,b show the AFM images of the *cTi* surface. Accordingly, the nanosized hemispherical patterns have a depth and diameter of 20 ± 5 and 100 ± 20 nm, respectively (Figure 5c).

### 2.3. Response of aTiO_2_ and cTi Surfaces to Plastic Deformation

Stents are subjected to considerable plastic deformation during application. This aspect is not generally considered in the studies aiming to modify the surfaces of the stents by different methods. Understanding the changes that may be induced on the structure and morphology of the surface-treated stents is very important for the proper functioning of the stents. We have applied a three-point bending test (for two different angles: 45° and 90°) to compare the mechanical stability of anodized samples before (*aTiO_2_*) and after (*cTi*) removal of the nanoporous oxide structure.

After the bending test, SEM images of the *aTiO_2_* surface indicated the formation of microcracks in the nanoporous structure (Figure 6a,b). However, the *cTi* surface survived well under both deformation conditions (Figure 6c,d). These results indicated that the samples with a nanoporous anodic oxide layer (*aTiO_2_*) tend to crack and delaminate. This effect may create debris during the functioning of the stents and increase the risk of increased augmented vascular inflammation. On the other hand, the nanopatterned surfaces (*cTi*) did not exhibit cracking or delamination under the bending conditions used in this study. Additionally, they preserved their nanopatterned morphology, making them more favorable for stent applications. Bending tests of plasma-treated *cTi* surfaces also afforded similar results. The potential of these samples to be used as stent materials is investigated in detail in the further sections of the study.

### 2.4. Surface Analysis by XPS

XPS characterization of *cTi* samples before and after the plasma treatment was performed to reveal surface chemistry (Figure 7). The wide spectrum obtained from sample surfaces before plasma treatment consisted of mainly titanium and oxygen, typical for a metallic titanium surface exposed to the atmosphere. In addition to carbon, nitrogen, and calcium probably originating from surface contamination, the peak corresponding to F1s was also present in the spectrum due to the electrochemical process used for fabrication of nanostructures. The spectrum acquired after plasma treatment was similar to the one obtained before plasma treatment except for the F1s peak, which was no longer present on the surface.

High-resolution XPS spectra of Ti2p, O1s, and F1s were also acquired to detail the surface’s chemistry and the effect of the oxygen plasma treatment. The deconvoluted spectrum of the Ti2p peak before oxygen plasma consisted of two doublets (Figure 8). The primary compound of the film was Ti^4+^ bound to oxygen in the form of TiO_2_, as expected (Ti2p_3/2_ = 458.7 eV, Δ = 5.8 eV). The minor compound in the film is attributed to fluorinated titanium [42] (Ti2p_3/2_ = 460.1 eV, Δ = 5 eV). The presence of the fluorinated compound was further verified with the F1s peak (Figure 9a). The deconvoluted O1s spectrum, which is the other principal constituent of the surface layer, consists of two compounds at 530.2 eV (O1s for TiO_2_) and 531.6 eV (O1s for adsorbed O–H or O–C molecules) [43] (Figure 9b). Thus, the surface layer of the sample before plasma treatment consists of TiO_2_-, fluorinated-Ti-, O–H-, and O–C-based compounds.

After oxygen plasma treatment, the fluorinated-Ti in the Ti2p spectrum was no longer observed, leaving behind TiO_2_ as the main titanium compound (Figure 8). The F1s peak was also no longer detectable in the spectral region of F1s (Figure 9a). The O1s intensity that corresponds to the Ti^4+^ bond to oxygen increased (Figure 9b) along with the substantial decrease of O1s peaks corresponding to O–H and O–C bonds.

For further evaluation of the concentration variation of Ti, O, F, and C within the surface layer, we performed XPS depth profiling on *cTi* samples before and after plasma treatment. According to the XPS depth profiles shown in Figure 10, plasma treatment caused an increase in the thickness of the oxide layer, from about 5 to 10 nm (orange and green arrows). In addition, the appearance of a fluorine peak in the depth profile after sputtering 5 nm of the surface layer indicates that the film grew above the existing oxide-fluoride mixture layer during oxygen plasma treatment (Figure 10b). Additionally, the calcium contamination observed in the wide spectrum was also diminished after a few cycles of sputtering.

### 2.5. Surface Wettability

Evaluation of surface wettability is one of the surface characteristics of biomaterial surfaces, as it may play an essential role in the biological response. Water contact angles (WCA) were measured on *pTi* and *cTi* samples before and after plasma treatment to determine the contribution of both nanopatterning and plasma treatment to wetting properties (Table 1).

Results show that the *pTi* and *cTi* are relatively hydrophobic. Plasma treatment increased the wettability of both samples due to the removal of surface contaminants and the oxidation of titanium, which was also confirmed by XPS analysis conducted on *cTi* (Figure 7, Figure 8 and Figure 9). However, plasma treatment induced a more prominent effect on *pTi*. 

Increased wetting properties of the surface do not necessarily indicate a better biological response [44], which also depends on surface texture, chemistry, and the nature of the cells that interact with the surface. Thus, measured WCA values should be considered together with other surface features for its final application in the body (desired cell interaction).

### 2.6. Biological Response of the Nanopatterned Surfaces

For the description of the nanopatterned surfaces that will be subjected to biocompatibility tests, the results of XPS, SEM, and surface wettability data for *cTi* and *cTi+P* are compiled and presented in this section.

The structure of the nanoporous/nanotubular titanium layer formed during anodization in fluoride-containing electrolytes is very well-known [45]. During anodization, a fluorinated-Ti layer forms below the barrier layer and between the pore walls (Figure 11a) [46]. As expressed by XPS measurements (Figure 6, Figure 7, Figure 8 and Figure 9), the remnants of this fluorinated-Ti layer were still present on the surface even after removing the anodic oxide with ultrasonication. Thus, the morphology and chemistry of the sample described as *cTi* is equivalent to the schematic picture presented in Figure 11b and consists of a mixture of fluorinated-Ti and oxides with a surface wetting angle of 80 ± 4°.

Oxygen plasma treatment increases the oxide layer thickness on metallic titanium. As determined from XPS depth profiles (Figure 10a,b), this oxide film grows on the mixture of fluorinated-Ti and oxides. Thus, on the *cTi+P* samples, the fluorinated-Ti layer was not present, and their wetting angles were lower than those of *cTi* (Figure 11c).

#### 2.6.1. In Vitro Cell Viability Studies

To reveal the influence of nanopatterning on the biological response, we performed in vitro studies with *HCAEC* and *HCASMC*. These two cell lines are often used in in vitro studies of cardiovascular functions, such as angiogenesis and atherosclerosis. *HCAEC* form a layer on the inner side of the blood vessels and have a main role in the formation of the vessels. *HCASMC* form the media layer of arteries and maintain the integrity of the arterial wall. In the case of atherosclerosis and restenosis, these cells play an important role in arterial wall remodeling. In case of vascular stents, it is desired that endothelial cells grow well on the surface, as they present ideal antithrombogenic material, while smooth muscle cell growth should be inhibited to prevent the risk of restenosis (narrowing of blood vessels).

In vitro tests were performed on *pTi* and *cTi* surfaces before and after plasma treatment to determine the metabolic activity of cells adhered to these substrates. *pTi* samples were used as a reference.

The *HCAEC* viability on the *cTi* samples was significantly improved compared to *pTi* reference samples (Figure 12a). The nanopatterned surface *cTi* covered with a mixture of titanium dioxide and fluorinated-Ti was shown to improve *HCAEC* viability, while even higher metabolic activity of adhered cells was observed on the plasma-treated nanopatterned surface (*cTi+P*) (Figure 11c).

On the other hand, *HCASMC* (Figure 12b) adhered and proliferated best on *pTi*, whereas on *cTi* samples, their viability was significantly lower, indicating unfavorable surface conditions for their attachment and viability. In the initial stage, plasma-treated nanopatterned surfaces (*cTi+P*) showed improved viability of *HCASMC* compared to nanostructured surfaces (*cTi*); however, after 24 h of incubation, no significant difference in viability was observed between these two surfaces. It should also be emphasized that *HCASMC* viability was the lowest for nanopatterned surfaces (with or without plasma treatment) compared to the control (*pTi*), which indicates that such nano-topography significantly influences the adhesion and viability of *HCASMC*. Interestingly, the opposite was observed for the *HCAEC*, where nanopatterned surfaces were shown to improve cell viability. After plasma treatment of nanopatterned surfaces and longer incubation times, the viability of *HCAEC* was improved.

#### 2.6.2. Platelet Adhesion

Platelet adhesion tests were utilized to determine the hemocompatibility of the *cTi* and *cTi+P* samples that exhibited good biocompatibility, using *pTi* as a reference. After incubation with whole blood, SEM analyses were used to determine the number and morphology of adhered platelets on the samples. The morphological forms of platelets from the least activated to the most activated were as follows: round (R) > dendritic (D) > spread dendritic (SD) > spread (S) > fully spread (FS) [47]. Figure 13 presents the interaction of whole blood with different samples. It was observed that on the *pTi* surface, platelets adhered mainly in fully spread form, which indicates their high activation on the surface and thus the high risk for undesired thrombotic reactions. Some erythrocytes were also present on the surface, as seen in Figure 13. In the case of plasma-treated Ti (*pTi+P*), platelets were mostly in the dendritic (D) and spread dendritic (SD) form. In this case, a lower number of platelets on the surface was observed, and some erythrocytes were also detected as well as leukocytes. In the case of *cTi*, a high number of platelets was observed, and their morphology indicated high activation and adhesion, as platelets were mainly in the S and FS form. While in the case of *cTi+P*, platelet adhesion was significantly reduced, and practically no platelets were detected on these surfaces, only a few erythrocytes were present, as seen in Figure 13. This result pointed out the benefit of oxygen plasma treatment on hemocompatibility.

## 3. Discussion

The beneficial role of nano-topographic features on the biocompatibility of biomaterials is very well-known and has already been the subject of intensive research [5,48,49,50,51,52,53,54,55,56]. Most of the studies deal with osseointegration properties of orthopedic implants, while some also involve the interaction of endothelial cells with the surface used for cardiovascular stent applications [48,50,51]. It was shown that surface geometry plays an important role in biological responses, as different interactions of cells and proteins with surfaces having similar chemistry but altered nanotopography were observed [17,18,19,34,50]. Accordingly, the biocompatibility of an implant with appropriate surface chemistry can be further increased with nano-structuring. The proposed mechanisms that lead to increased biocompatibility are: (1) electronic modifications of the topmost surface, (2) increased surface area (edges/corner sites, particle boundaries), and (3) mimicking the natural architecture of vascular walls [52]. These mechanisms potentially affect protein adhesion (their conformation) and the cell response, which further influences its integration with the surrounding tissues and the lifespan of the medical device in the body.

The nanopatterned surfaces obtained in this study were shown to provide selective cell adhesion, as improved proliferation of *HCAEC* and reduced proliferation of *HCASMC* were observed. In addition, these types of surfaces seem to reduce platelet adhesion and activation, which reduces the risk of thrombosis. Other studies show that the viability of both *HCAEC* and *HCASMC* increases without selectivity for nanopore diameters below 35 nm [15]. However, for those between 70 and 100 nm, selectivity toward *HCAEC* was obtained [15,57]. We have observed similar vascular cell responses using our nanopatterned surfaces with 100 ± 10 nm pore diameters. The higher cell viability of *HCAEC* compared to *HCASMC* (Figure 12) can be attributed to the morphology of the obtained nanopattern that resembles the natural nanoarchitecture of the internal vessel wall where endothelial cells attach [58], as well as to altered surface chemistry obtained by oxygen plasma treatment.

Fluorinated titanium compounds were detected on nanopatterned surfaces after removing nanoporous TiO_2_ with ultrasonication (Figure 8, Figure 9 and Figure 10). However, further improvement of cell viability with oxygen plasma treatment was achieved for *HCAEC*, while no significant difference in cell viability was observed for *HCASMC*. It could be proposed that *HCAEC* are more sensitive to the change in surface chemistry, as after plasma treatment fluorine is removed from the surface and the oxide layer is formed. However, our results also show that the fluorinated-Ti-containing nanopatterned (*cTi*) surface exhibits better cell viability for *HCAEC* than flat (*pTi*) and (*pTi+P*), indicating the dominant role of nanopatterning on cell viability even in the presence of a fluorinated layer.

The nanofeatures obtained in this study have significantly lower thicknesses than nanotubes described in the literature. These results indicate that the main morphological parameter to cell viability is the diameter of the nanofeatures rather than its thickness/length. Thus, the use of poorly adherent thick coatings, such as TiO_2_ nanotubes/nanopores, to improve the cell viability of implants is not mandatory. As proposed, nanopatterned metallic titanium with similar diameters covered by a very thin oxide layer can function similarly.

Attaining the desired biocompatibility of stents is not sufficient, as stents should also have good hemocompatibility. Primarily, they should prevent adhesion and activation of platelets on the surface, as this would significantly reduce the risk for thrombotic reactions [32]. Among the samples investigated in this study, the desired hemocompatibility properties were only attained for oxygen plasma-treated nanopatterned samples (*cTi+P*) (Figure 13). It is well-known that titanium can act as a thrombogenic material depending on its surface properties [59]. Several studies in the literature indicate the positive role of fluorination on the osseointegration and thrombogenic properties of titanium-based orthopedic and dental implants [60]. Therefore, fluorides may induce thrombogenic reactions on the surface, while oxygen plasma treatment can overcome this issue as it removes the fluoride from the surface as well as increases the concentration of oxygen on the surface. We have not come across any studies on the possible role of fluorides on hemocompatibility and cell viability for stent applications, and further work is needed to clarify these effects.

To maintain a healthy blood vessel and avoid *ISR*, the endothelization of the inner layer of the vessel is very important for cardiovascular stent applications. As the preferred proliferation of *HCASMC* is one of the main reasons for *ISR*, providing a surface feature that offers improved cell selectivity towards *HCAEC* is desired. The proposed nanopatterning method obtained promising results in this regard. However, hemocompatibility is also crucial for the proper functioning of the stent material; thus, appropriate surface chemistry is also needed (restricting the contact with fluorinated-Ti, increasing oxygen content, etc.). Oxygen plasma treatment used in this study was successful for this purpose. An additional benefit of the proposed method is the mechanical integrity of the nanopatterned layer on the stent material under plastic deformation, as cracking and spallation of the oxide layer may lead to undesired biological responses after implantation of the stent and its long-term use in the human body.

## 4. Materials and Methods

### 4.1. Sample Preparation

The sample preparation procedure schematically presented in Figure 1 is detailed in the following sections.

#### 4.1.1. (STEP #1) Ti Coating

The 0.30 mm-thick 316 LVM stainless-steel samples were cut to rectangular pieces of 20 × 10 mm dimensions and mirror-polished. They were placed in a vacuum chamber after surface cleaning with acetone, ethanol, and distilled water. After evacuation to a base pressure of 10^−3^ Pa, 900 sccm Ar gas was introduced into the chamber to reach 1.5 Pa. Argon ions first etched the samples. The ion etching process was started by applying −600 V bias voltage against the grounded chamber for 4 min, followed by −800 V for 2 min and −1000 V for 1 min. 

After ion etching, samples were coated with titanium using a dual-rotating magnetron sputtering source operating at 40 kHz, 550 V, and 18 A at a pressure of 0.5 Pa Argon (200 sccm) in the same experimental system, without breaking the vacuum conditions. During the deposition, a DC voltage of −150 V was applied to the substrates to ensure the appropriate properties of the titanium film. The deposition was accomplished in 500 s. The thickness of the titanium film on the stainless-steel substrate was 1 µm. These samples are referred to as plain-titanium-coated stainless-steel (*pTi*).

#### 4.1.2. (STEP #2) Anodization

The plain-titanium-coated SS samples (*pTi*) were anodized in ethylene glycol solution containing 0.6 wt.% NH_4_F and 1% *v*/*v* H_2_O at 27 ± 1 °C. A water-jacketed 100 mL cell was used to precisely control the temperature of the anodization solution. The distance between the stainless-steel cathode and the sample was kept constant at 20 mm. The anodization voltage was 40 V. The duration of anodization was optimized by following the time-dependent current variation (Figure 2). After accomplishing the anodization, samples are washed with distilled water and dried. These samples are named as anodized titanium oxide (*aTiO_2_*).

#### 4.1.3. (STEP #3) Ultrasonication (US)

We have benefited from the low adhesion of anodized titanium oxide (*aTiO_2_*) to metallic titanium to remove the *aTiO_2_* layer from the titanium surface. For this purpose, we have subjected the *aTiO_2_* to ultrasonication using a horn-tip ultrasonicator in a distilled water bath, with a peak-to-peak amplitude of 25 µm (transversal mechanical oscillation) and a frequency of 20 kHz. Details of the device are presented elsewhere [61]. The samples subjected to US to produce the nanopatterned metallic titanium layer are named as *cTi*.

#### 4.1.4. (STEP #4) Plasma Treatment

After the ultrasonication, a series of samples were treated in oxygen plasma to observe the contribution to surface chemistry and biocompatibility. This step aims to activate the sample surface and remove any organic impurities that might have persisted on the sample’s surface after accomplishing the ultrasonication. We used a radiofrequency (RF) generator operating at 13.56 MHz and a peak-to-peak voltage of 600 V. The generator was connected to a coil via a matching network. The nominal power was fixed at 400 W, and the plasma was ignited in the capacitive mode (E-mode). Samples were treated in oxygen plasma under 50 Pa pressure for 10 s. Plasma-treated, nanopatterned, metallic titanium and plasma-treated plain titanium are named as *cTi+P* and *pTi+P*, respectively.

### 4.2. Sample Characterization

#### 4.2.1. Surface Morphology and Coating Thickness Determination

The surface morphology of the coatings after each step was investigated with FEG-SEM (JEOL JSM 7000F and Thermo Scientific Quattro S Field Emission Scanning Electron Microscope). Topographic features of the surfaces after removal of the oxide layer were examined by Atomic Force Microscopy (Nanomagnetics Instruments, Ankara, Turkey) in tapping mode in the air using Si cantilever at a constant force of 30 N/m and resonance frequency of 160 kHz (<10 nm tip radius, 10 µm tip height).

The thickness of the metallic titanium coatings on the 316 LVM substrates was measured with an XRF thickness measurement system (Fischerscope X-ray System XDL 662). The device was calibrated for titanium coatings on 316 LVM stainless-steel.

#### 4.2.2. XPS Analysis

X-ray photoelectron spectroscopy (XPS) analyses were performed with the PHI-TFA XPS spectrometer (Physical Electronics Inc., Chanhassen, MN, USA) equipped with an Al-Kα monochromatic source. The analyzed area was about 0.4 mm in diameter. The high-energy resolution spectra were acquired with an energy analyzer operating at the resolution of 0.6 eV and pass energy of 29 eV. XPS depth profiles were obtained by sputtering the sample surface with a 4 keV beam of Ar ions rastered over 3 × 3 mm. The sputtering rate was determined using standards and was about 1 nm/min. Quantification of surface composition was performed from XPS peak intensities considering relative sensitivity factors provided by the instrument manufacturer [62].

#### 4.2.3. Bending Tests

We applied a three-point bending test to determine the SS samples’ response to plastic deformation covered with *aTiO_2_* and *cTi*. The samples were bent at 45° and 90°. After bending tests, the surfaces of all samples were investigated with SEM.

#### 4.2.4. Surface Wettability

The surface wettability was performed with the Drop Shape Analyzer DSA-100 (Krüss GmbH, Hannover, Germany) by the sessile drop method to measure the static contact angle. The contact angle on the surface was analyzed immediately after plasma treatment by adding a 2.5 µL drop of deionized water on 8 different surface areas. The relative humidity was around 45%, and the operating temperature was 21 °C, which did not vary significantly during continuous measurements.

### 4.3. In Vitro Biological Response

Human coronary artery endothelial cells (*HCAEC*) and human coronary artery smooth muscle cells (*HCASMC*) were used to evaluate the interaction of cells with the surface. Cell viability was determined via the MTT assay. The blood compatibility was assessed by the adhesion and activation of platelets on the surface after incubation with whole blood. All the samples were incubated with biological material immediately after plasma treatment because of the instability of the WCA of plasma-treated samples after exposure to atmosphere [63].

#### 4.3.1. Cell Viability Studies—MTT Assay

*HCAEC* were purchased from Lifeline Cell Technology (Frederick, MD, USA), and *HCASMC* were purchased from ProVitro AG (Berlin, Germany). *HCAEC* were grown in the VascuLife EnGS endothelial medium complete kit (Frederick, MD, USA) and *HCASMC* in the smooth muscle cell growth medium FCS-kit (ProVitro AG, Berlin, Germany) at 37 °C in a humidified atmosphere at 5% CO_2_. Cells were seeded at a density of 2 × 10^4^ cells in 100 μL drops of the medium on the upper side of the samples (concentration: 2.55 × 10^4^ cells/cm^2^) and left for 3, 24, and 72 h to attach. Experiments were performed in triplicates for each sample and incubation time.

After 3, 24, and 72 h of the *HCASMC* and *HCAEC* incubation process, the medium was removed, and the samples were washed to remove all the unattached cells from the surface. Then, 200 μL of fresh Hanks’ Balanced Salt Solution (*HBSS*) mixed with the tetrazolium agent was added to each well with the sample, mixed, and the OD at 492 nm was measured with a microplate reader (Tecan Infinite M1000). Blank (medium without the cells) was measured at 690 nm, and all the experiments were performed in triplicates.

JASP 0.9.2 open-source software (University of Amsterdam, Amsterdam, The Netherlands) was used to statistically analyze MTT data. Group means were calculated and compared using ANOVA followed by the post hoc Tukey’s range test. Differences in the means were considered statistically significant at *p* < 0.05. Error bars in charts represent standard error.

#### 4.3.2. Platelet Adhesion

Incubation with whole blood was performed to study the adhesion and activation of platelets on the nanopatterned Ti surface (*cTi*) and nanopatterned plasma-treated surface (*cTi+P*) using *pTi* and *pTi+P* as reference samples. Whole blood was obtained from healthy donors by vein puncture. Samples were cut to 7 × 7 mm pieces and incubated with 250 μL of whole blood for 45 min at room temperature. Afterward, 250 μL of PBS was added and rinsed 3 times with PBS to remove weakly adherent platelets. The adherent cells were fixed by 400 μL of 0.5 vol% glutaraldehyde for 1 h at room temperature. For SEM analysis, the surfaces were again rinsed with PBS and dehydrated by using a graded ethanol series (50, 70, 80, 90, 100 vol%), and again 100 vol% of ethanol for 5 min, and in the last stage, 100 vol% of ethanol for 10 min. Afterward, the samples were dried with liquid nitrogen, left in a vacuum for 3 h, and coated with gold/palladium before SEM characterization.

## 5. Conclusions

A method for synthesizing titanium coatings on stainless-steel surfaces with desired morphology and chemistry for stent applications was presented. The method enables the formation of periodical, nanopatterned, thin titanium film on smooth stainless-steel substrates. The unique morphology and chemistry were obtained by the four-step procedure, where titanium film was first deposited on the stainless-steel surface, followed by anodization, ultrasonication, and oxygen plasma treatment. The anodization enabled the formation of nanoporous/nanotubular anodic oxide consisting of titanium oxide and fluorinated titanium. After removing this fragile oxide layer by ultrasonication, a nanopatterned titanium surface with periodical structures of about 20 nm in depth and about 100 nm in diameter was obtained. The diameter of the nanopatterns is in accordance with the sizes of biocompatible nanoporous titanium oxides reported in the literature. When subjected to plastic deformation, these structures retained their shape and integrity compared to the nanoporous oxide-covered surfaces. The results indicate that our unique surface preparation process substantially decreased the cracking and spalling tendency of the surface layer during the implantation of stents. Such a surface morphology in combination with altered surface chemistry and wettability was also beneficial for the adhesion and proliferation of *HCAEC*.

On the other hand, the adhesion and proliferation of *HCASMC* were suppressed mainly due to the altered surface nano-topography. The rapid endothelialization and suppressing the proliferation of smooth muscle cells are crucial for the performance of vascular stents since they prevent thrombosis and restenosis. Furthermore, the activation of blood platelets on the nanopatterned surface was significantly suppressed after oxygen plasma treatment. This effect indicated the role of additional surface features (chemistry and wettability) induced by oxygen plasma as the nanopatterned surface alone did not show a reduction in the number of adhered platelets.

## Figures and Tables

**Figure 1 ijms-23-04595-f001:**
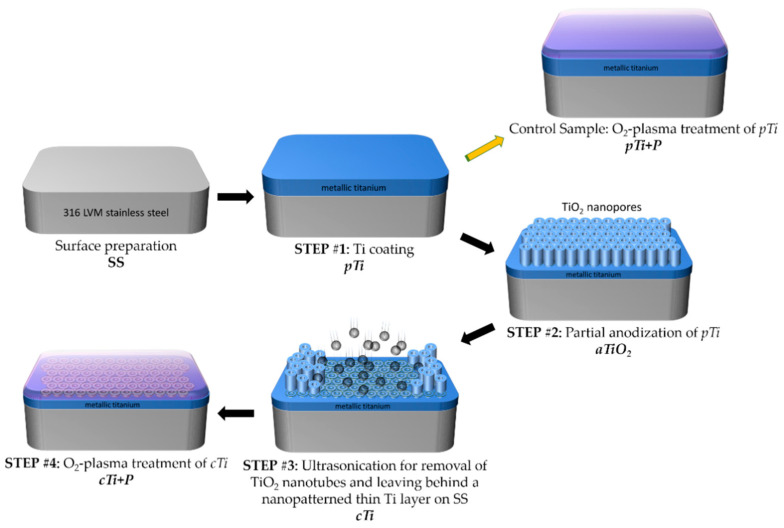
Surface preparation steps: #1 titanium coating on stainless-steel, #2 anodization of titanium, #3 ultrasonication of nanoporous TiO_2_, and #4 O_2_-plasma treatment of *cTi*.

**Figure 2 ijms-23-04595-f002:**
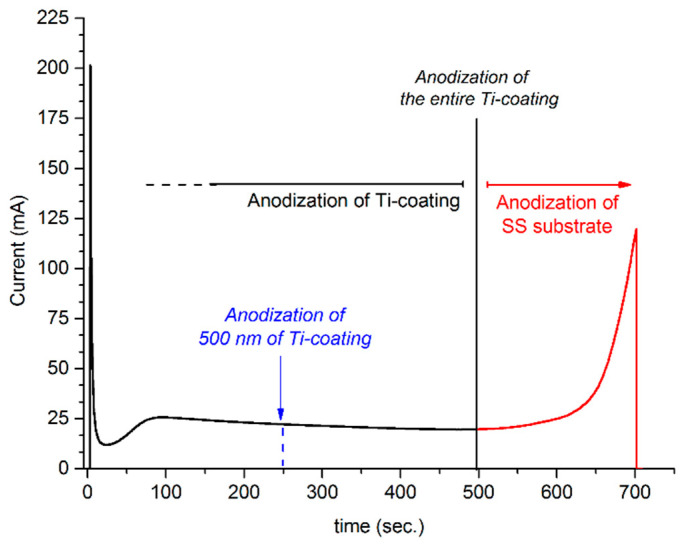
Time dependence of the anodization current.

**Figure 3 ijms-23-04595-f003:**
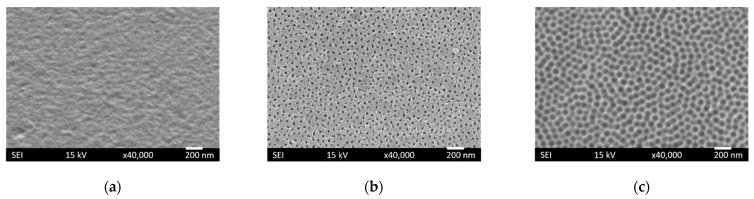
Surface morphology from SEM images of samples after each step: (**a**) as-deposited (*pTi*), (**b**) as-anodized (*aTiO_2_*), and (**c**) as-ultrasonicated (*cTi*).

**Figure 4 ijms-23-04595-f004:**
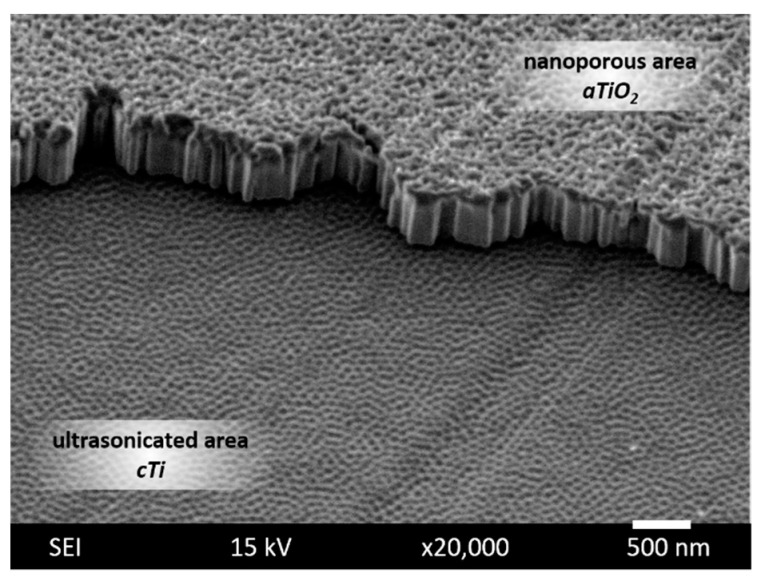
Surface morphology of nanoporous (*aTiO_2_*) and ultrasonicated (*cTi*) areas from SEM analysis. The sample was tilted at 30°.

**Figure 5 ijms-23-04595-f005:**
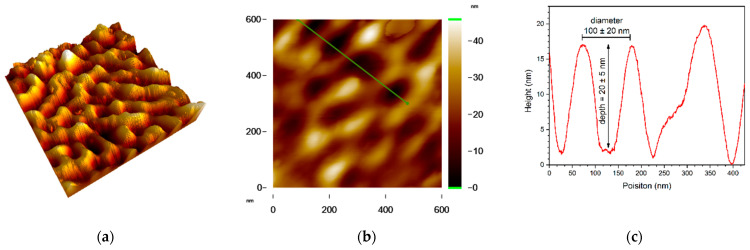
AFM images of the nanopatterned titanium surface (*cTi*): (**a**) 3D image of 1 × 1 µm area, (**b**) 2D image of 600 × 600 nm area, and (**c**) line profile taken from the region marked with red line on (**b**).

**Figure 6 ijms-23-04595-f006:**
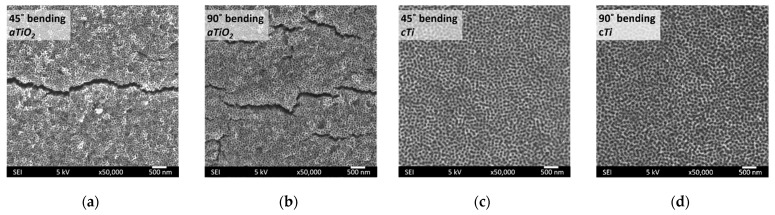
SEM images of the surface after the three-point bending tests: (**a**) 45° bending of *aTiO_2_*, (**b**) 90° bending of *aTiO_2_*, (**c**) 45° bending of *cTi*, and (**d**) 90° bending of *cTi*.

**Figure 7 ijms-23-04595-f007:**
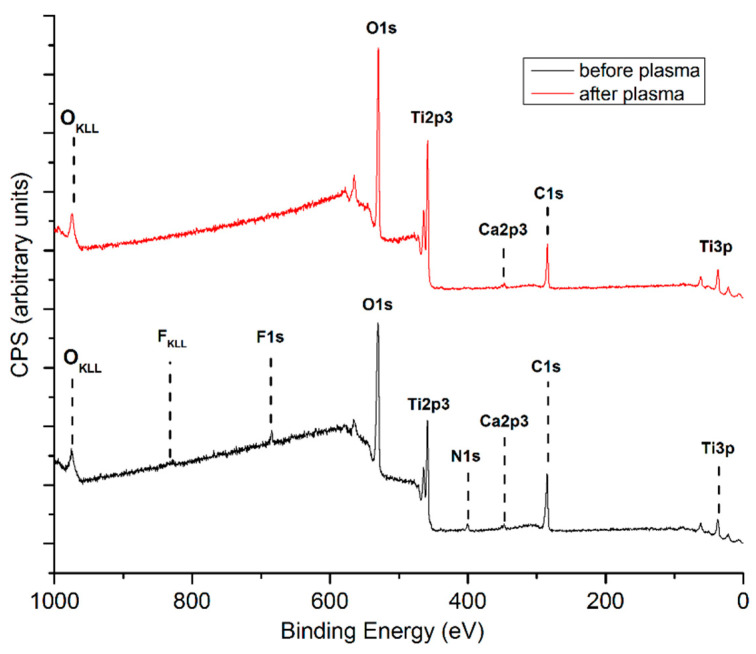
Wide XPS spectra of *cTi* before and after oxygen plasma treatment.

**Figure 8 ijms-23-04595-f008:**
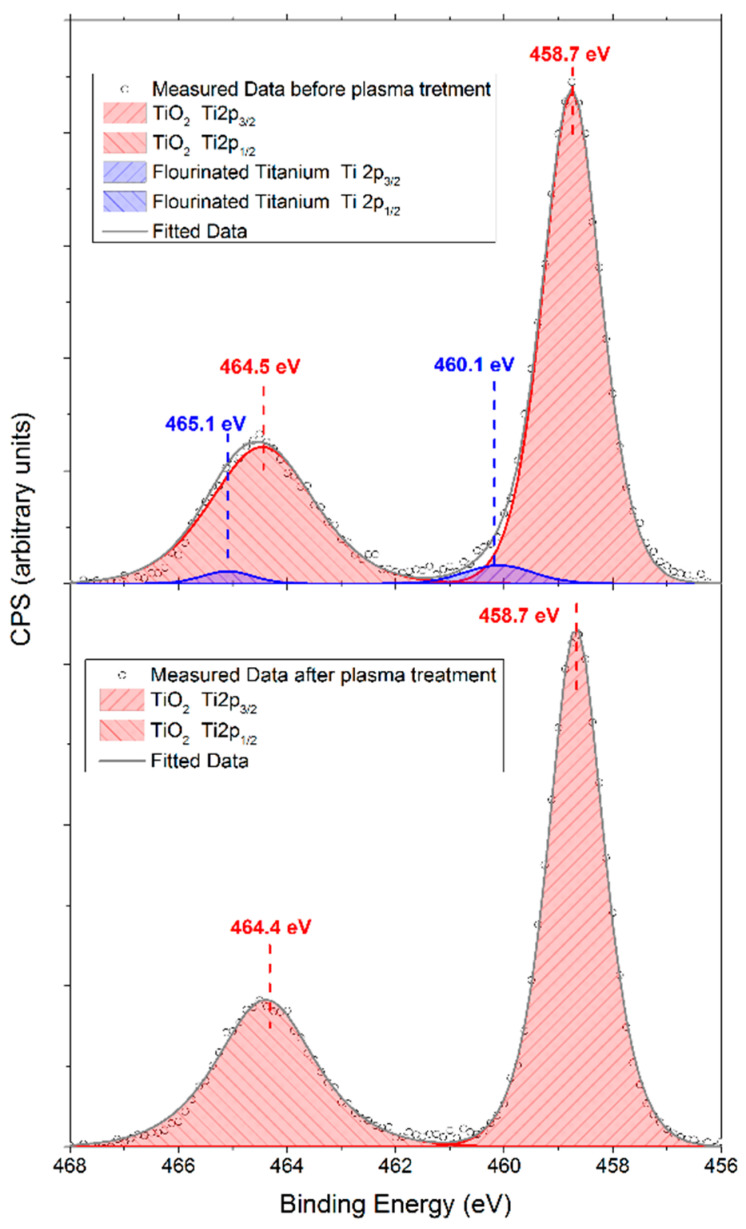
XPS spectra of Ti2p of *cTi* before and after oxygen plasma treatment.

**Figure 9 ijms-23-04595-f009:**
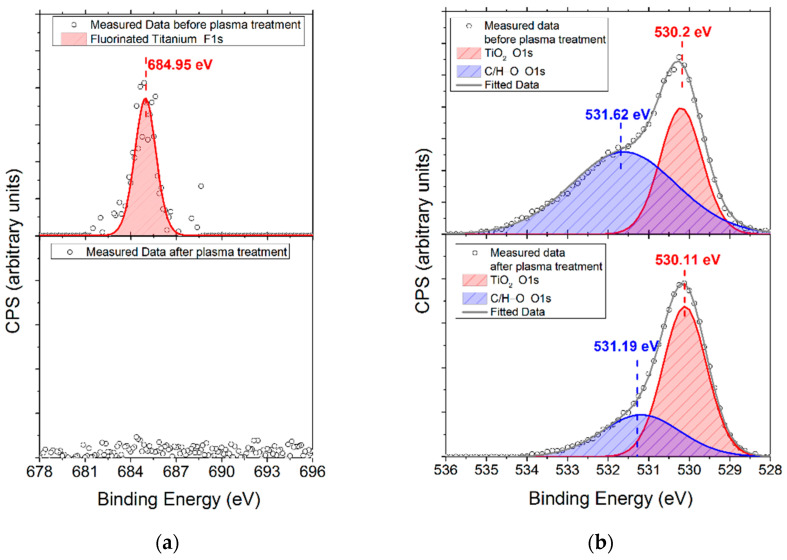
XPS spectra of (**a**) F1s and (**b**) O1s regions of *cTi* before and after oxygen plasma treatment.

**Figure 10 ijms-23-04595-f010:**
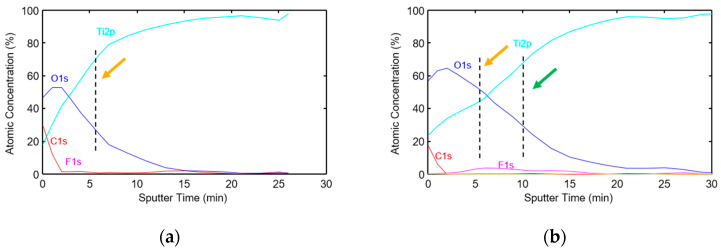
XPS depth profile of *cTi* (**a**) before and (**b**) after oxygen plasma treatment (orange and green arrows represent oxide layer thickness before and after the plasma treatment, respectively).

**Figure 11 ijms-23-04595-f011:**
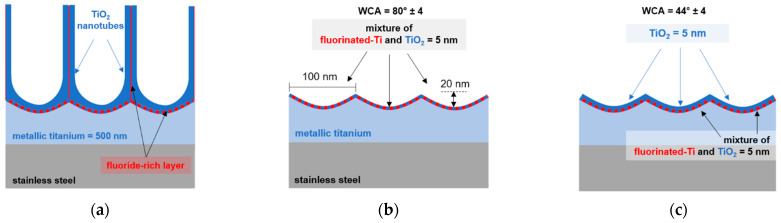
Schematic representation of the surface chemistry and morphology of (**a**) *aTiO_2_*, (**b**) *cTi*, and (**c**) *cTi+P* surfaces.

**Figure 12 ijms-23-04595-f012:**
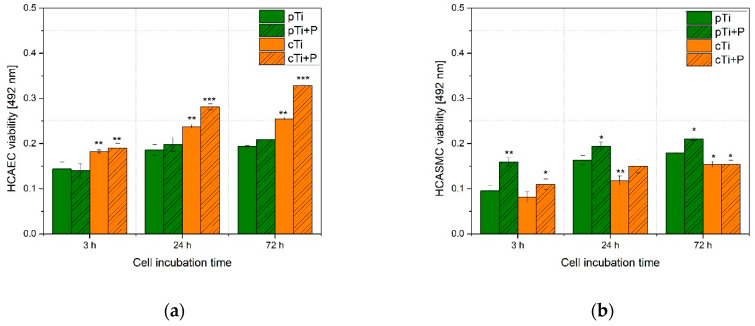
Results of the MTT assay for: (**a**) *HCAEC* and (**b**) *HCASMC* cell proliferation on *pTi*, *cTi*, *pTi+P,* and *cTi+P* surfaces. The error bars represents standard error. Symbols * represent a statistically significant change in viability after 3, 24, and 72 h (represents statistical significance at * *p* < 0.05, ** *p* < 0.01, and *** *p* < 0.001 compared with the *pTi*). Mean values (±SE) for the respective triplicates are presented.

**Figure 13 ijms-23-04595-f013:**
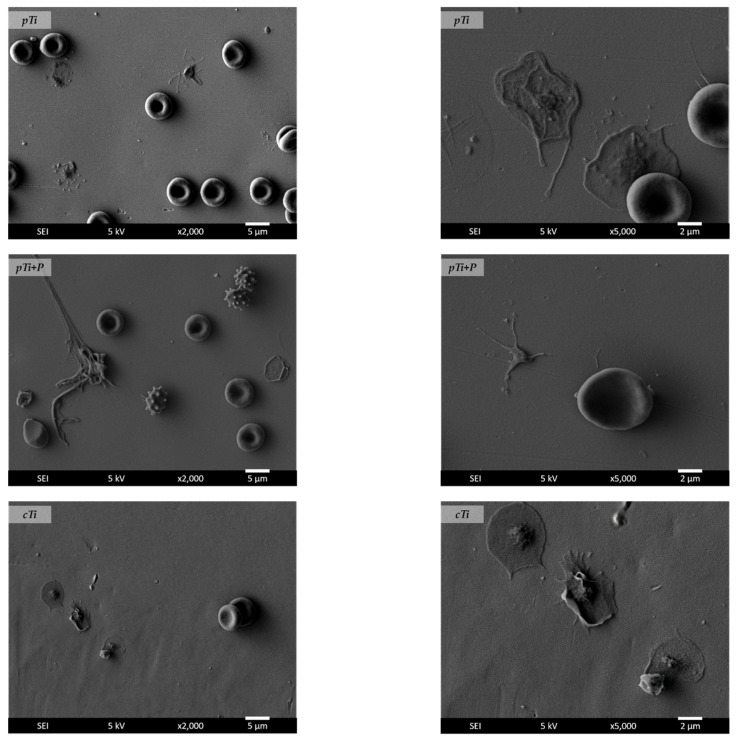
Platelets interacting with plain titanium (*pTi*), plain titanium plasma-treated (*pTi+P*), nanopatterned Ti surface (*cTi*), and nanopatterned plasma-treated surface (*cTi+P*) at different magnifications (left-hand side 2 kX, right-hand side 5 kX). Images were obtained from SEM analysis of samples incubated with whole blood.

**Table 1 ijms-23-04595-t001:** WCA values before and after plasma treatment.

*pTi*	*cTi*	*pTi*+*P*	*cTi*+P
85 ± 3°	80 ± 4°	28 ± 2°	44 ± 3°

## Data Availability

Not applicable.

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
