# Peer review of "Biocompatibility and Mechanical Stability of Nanopatterned Titanium Films on Stainless Steel Vascular Stents"

_ijms, 2022, doi:10.3390/ijms23094595_

Round 1
Reviewer 1 Report
Overall the study is very interesting, however there are some aspects of the manuscript that need to be critically addressed.
- The Abstract needs significant revision with the following:
- One or two statements regarding background and research motivation
- The abstract currently reads as a Methodology section, instead of an overview of the background, motivation, method, results and significance. These various aspects need to be included
- Significance of the results and novelty of the study need to be communicated.
- What wider application does this work have, how can it be used in the medical industry?
- Some important references missing regarding TiO2 nanostructured surfaces, e.g.
- Mechanical, bactericidal and osteogenic behaviours of hydrothermally synthesised TiO2 nanowire arrays (https://doi.org/10.1016/j.jmbbm.2018.02.011)
- Bacteria Death and Osteoblast Metabolic Activity Correlated to Hydrothermally Synthesised TiO2 Surface Properties (https://doi.org/10.3390/molecules24071201)
- There is a significant lack of references in Section 2. Results and Discussion. While there is a large amount of quality Results presented in this section, the manuscript fails to explain why these results have occurred, how they are related to nanostructures and what is the mechanism behind these results. These are critical in a Discussion and need to be included, along with published references that support the authors’ claims.
Reviewer 2 Report
The presented manuscript discusses about fabrication of nanopatterned titanium films on stainless steel vascular stents to improve biocompatibility. The manuscript seems to present interesting results and can be published in the journal after some revisions as below:
- The English language and structure should be improved
- The authors should discuss about the toxic effects of alloying elements in stainless steel and also fluoride due to importance of this issue in biomedical applications. They can refer to below papers.
- The effects of fluoride on bone and implant histomorphometry in growing rats, https://doi.org/10.1002/jbmr.5650040405
- Fluoride Coatings on Magnesium Alloy Implants, https://doi.org/10.1155/2022/7636482
- The toxicity phenomenon and the related occurrence in metal and metal oxide nanoparticles: a brief review from the biomedical perspective, https://doi.org/10.3389/fbioe.2020.00822
- It seems that the introduction should be further expanded about the surface engineering of biomedical implants, authors can refer to below papers.
- What is the elastic modulus of this implant? Is it suitable for this application, please clarify this issue by some comparisons?
- It is suggested to add a schematic showing how this surface design can improve biocompatibility and other properties.
- Is there any mechanism that can be explained the effect of this surface engineering.
- Is there any hint about the response of surface against bacterial species?
Round 2
Reviewer 1 Report
The authors have addressed most of my previous concerns, including improving the Abstract and the addition of a Discussion section. The level of English in both of these sections should be improved.
While the Discussion contains some information, it seems very segregated in explaining the different results, instead of forming one cohesive picture of the results and mechanisms. While there is the addition of a number of references, more work is needed to have these references support the conclusions made in the Discussion, instead of highlight what other studies have done.
Overall there is some improvement, but a these couple of things require some more attention.
Author Response
Response to reviewer's comments:
1.) The authors have addressed most of my previous concerns, including improving the Abstract and the addition of a Discussion section. The level of English in both of these sections should be improved.
While the Discussion contains some information, it seems very segregated in explaining the different results, instead of forming one cohesive picture of the results and mechanisms. While there is the addition of a number of references, more work is needed to have these references support the conclusions made in the Discussion, instead of highlight what other studies have done.
Overall there is some improvement, but a these couple of things require some more attention.
Response: Thank you for your constructive comments.
We have made modifications on the discussion section to convert it into a more cohesive structure as you have suggested and supported our results with relevant references. We have further checked the English usage in the abstract and discussion section and made the necessary revisions to them. All the modified sections are highlighted in green.
Kind regards,
Ita Junkar
Reviewer 2 Report
The manuscript seems to be Ok and can be published in the present form, please do not forget to include all the mentioned papers.
Good luck
Author Response
Response to reviewer's comments:
1.) The authors have addressed most of my previous concerns, including improving the Abstract and the addition of a Discussion section. The level of English in both of these sections should be improved.
While the Discussion contains some information, it seems very segregated in explaining the different results, instead of forming one cohesive picture of the results and mechanisms. While there is the addition of a number of references, more work is needed to have these references support the conclusions made in the Discussion, instead of highlight what other studies have done.
Overall there is some improvement, but a these couple of things require some more attention.
Response: Thank you for your constructive comments.
We have made modifications on the discussion section to convert it into a more cohesive structure as you have suggested and supported our results with relevant references. We have further checked the English usage in the abstract and discussion section and made the necessary revisions to them. Please see the attached file of the modified manuscript, all the modified sections are highlighted in green.
Kind regards,
Ita Junkar